# LC-MS/MS Application in Pharmacotoxicological Field: Current State and New Applications

**DOI:** 10.3390/molecules28052127

**Published:** 2023-02-24

**Authors:** Cristian D’Ovidio, Marcello Locatelli, Miryam Perrucci, Luigi Ciriolo, Kenneth G. Furton, Isil Gazioglu, Abuzar Kabir, Giuseppe Maria Merone, Ugo de Grazia, Imran Ali, Antonio Maria Catena, Michele Treglia, Luigi T. Marsella, Fabio Savini

**Affiliations:** 1Department of Medicine and Aging Sciences, Section of Legal Medicine, University of Chieti–Pescara “G. d’Annunzio”, 66100 Chieti, Italy; 2Department of Pharmacy, University of Chieti–Pescara “G. d’Annunzio”, 66100 Chieti, Italy; 3International Forensic Research Institute, Department of Chemistry and Biochemistry, Florida International University, Miami, FL 33199, USA; 4Department of Analytical Chemistry, Faculty of Pharmacy, Bezmialem Vakif University, Fatih, Istanbul 34093, Turkey; 5Department of Pharmacy, Faculty of Allied Health Science, Daffodil International University, Dhaka 1216, Bangladesh; 6Pharmatoxicology Laboratory—Hospital “Santo Spirito”, 65124 Pescara, Italy; 7Laboratory of Neurological Biochemistry and Neuropharmacology, Fondazione IRCCS Istituto Neurologico Carlo Besta, 20133 Milan, Italy; 8Department of Chemistry, Jamia Millia Islamia (Central University), New Delhi 110025, India; 9Institute of Legal Medicine, University of Rome 2 “Tor Vergata”, 00133 Rome, Italy

**Keywords:** LC-MS/MS, illicit drugs, pharmacotoxicology, quantitative analyses, drugs, forensic applications

## Abstract

Nowadays, it is vital to have new, complete, and rapid methods to screen and follow pharmacotoxicological and forensic cases. In this context, an important role is undoubtedly played by liquid chromatography-tandem mass spectrometry (LC-MS/MS) thanks to its advanced features. This instrument configuration can offer comprehensive and complete analysis and is a very potent analytical tool in the hands of analysts for the correct identification and quantification of analytes. The present review paper discusses the applications of LC-MS/MS in pharmacotoxicological cases because it is impossible to ignore the importance of this powerful instrument for the rapid development of pharmacological and forensic advanced research in recent years. On one hand, pharmacology is fundamental for drug monitoring and helping people to find the so-called “personal therapy” or “personalized therapy”. On the other hand, toxicological and forensic LC-MS/MS represents the most critical instrument configuration applied to the screening and research of drugs and illicit drugs, giving critical support to law enforcement. Often the two areas are stackable, and for this reason, many methods include analytes attributable to both fields of application. In this manuscript, drugs and illicit drugs were divided in separate sections, with particular attention paid in the first section to therapeutic drug monitoring (TDM) and clinical approaches with a focus on central nervous system (CNS). The second section is focused on the methods developed in recent years for the determination of illicit drugs, often in combination with CNS drugs. All references considered herein cover the last 3 years, except for some specific and peculiar applications for which some more dated but still recent articles have been considered.

## 1. Introduction

This review aims to analyze the applications of liquid chromatography combined with mass spectrometry (LC-MS) in honor of the inventor of this powerful analytical instrument, Professor Gérard Hopfgartner. Analytical chemistry has a crucial role in preclinical and clinical studies, primarily regarding deaths and toxicity, because of its capability to develop accurate (precise and true) methods that allow for the quantification of drugs and illicit drugs from different biological matrices (both conventional and non-conventional) [1] with high level of confidence.

The use of liquid chromatography (LC) is growing rapidly, especially in pharmaceutical industries in their research and development studies. Particular attention is devoted to the instrument configuration that combines LC and mass spectrometry (MS) because, in this way, the central figures of merit related to an analytical method can be achieved (selectivity, sensitivity from the detector chosen, MS, and separation from LC) [2]. Thanks to high sensitivity and selectivity, many studies have reported liquid chromatography-tandem mass spectrometry (LC-MS/MS) as the primary analytical instrument [1,3]. In pharmacotoxicology, LC-MS/MS is used as the “gold” choice, even if in some cases it is not exchangeable with other types of instrumentations [4]. Therefore, it assures excellent versatility and, even though trained personnel are required, reduced analysis time or low spending resources are still significant benefits [5]. In this scenario, current laws continue to evolve, and the procedures and analytical methods must remain up to date [3]. Pharmacotoxicology coupled with analytical chemistry can give a sort of “instant photography” of the situations around us, from screening procedures to quantitative applications in forensic toxicology. It is imperative to have a wide range of information to prevent lawlessness, overdose deaths, and other avoidable unwanted situations [5]. Only analytical chemists can develop methods to determine different drugs starting from different and complex matrices such as whole blood, urine, plasma, saliva, etc., obtaining maximum advantages from the instrumentation in terms of sensitivity, selectivity, reproducibility, and ruggedness. Based on quantitative analysis, analytical chemistry can unravel forensic cases, as has been overwhelmingly evident in recent years [1]. As Seger reported in his manuscript, minor issues can be surmountable by studies, innovations, and continued research because LC-MS/MS needs particular attention or trained personnel [4].

The present review paper discusses the applications of LC-MS/MS in pharmacotoxicological cases because it is impossible to ignore the importance of this powerful instrument in the rapid development of pharmacological and forensic advanced research in recent years. On one hand, pharmacology is fundamental for drug monitoring and helping people to find the so-called “personal therapy” or “personalized therapy”. On the other hand, toxicological and forensic LC-MS/MS represents the most critical instrument configuration applied for drugs and illicit drugs screening and research, giving valuable support to law enforcement. Often the two areas are stackable, and for this reason, many methods include analytes attributable to both fields of application. In this manuscript, drugs were divided in separate sections. Particular consideration is given in the first section to therapeutic drug monitoring (TDM) and clinical approaches generally applied in pharmaceutical studies with a focus on the central nervous system (CNS), while the second section is focused on the methods developed in recent years for the determination of illicit drugs, often in combination with CNS drugs. All references considered herein cover the last 3 years, except for some specific and peculiar applications for which some more dated but still recent articles have been considered.

## 2. Drugs

Frequently, LC-MS/MS plays a vital role in pharmacokinetics (PK) and pharmacodynamics (PD) studies. Thanks to research progress, people with different types of cancer can take oral antineoplastic drugs. These oral drugs enable patients to avoid hospitalization, allowing a reduction in care costs. The main drawbacks are related to the fact that they must be able to adhere to the prescribed therapy. Additionally, some oral antineoplastic drugs have particular characteristics of pharmacokinetics. These shortcomings become important following individual therapeutic drug monitoring (TDM), aiming to avoid sub or toxic drug concentrations [6]. In TDM, the application of LC-MS/MS is one of the most used technologies, far surpassing older ones [4]. In reference to these goals, Llopis et al. (2021) developed a rapid method to estimate nine kinase inhibitors, two metabolites of them, and two antiandrogen drugs used for different types of cancer. Indeed, cobimetinib, dasatinib, ibrutinib, imatinib, nilotinib, palbociclib, ruxolitinib, sorafenib, and vemurafenib (kinase inhibitors) are used mainly for the treatment of hematological cancer and solid gastrointestinal tumors, but were also administered for the treatment of renal cell carcinoma and hepatocellular carcinoma. Abiraterone acetate and enzalutamide, antiandrogenic drugs, were approved in clinical practice for metastatic prostate cancer treatment. The method was immediate and very fast because it needed just 2.8 min, with a non-linear mobile phase gradient. After a single step related to the sample pre-treatment, specifically protein precipitation (PP) associated with the selected matrix (plasma), 10 µL was directly injected into LC-MS/MS instrumentation [7].

Ferrari et al. validated a “quick and robust LC-MS/MS method” to quantify four antibiotics: piperacillin, meropenem, linezolid, and teicoplanin. This choice came from research about hospitalized patients and their antibiotic treatment. Eighty plasma samples from 49 patients were considered and pre-treated with liquid–liquid extraction (LLE), and 5 μL was analyzed employing an LC gradient elution. The mobile phases were, respectively, water and methanol (both with 0.1% formic acid to improve the ionization efficiency allowing an improvement of the instrumental method sensitivity). Mobile phases play an important role during the analysis because they immediately reverse to finish as starting conditions. In addition, in this case, the group highlighted the importance of personalized therapy following the specific characteristics of every single patient [8] following the concept of the so-called “personal therapy” or “personalized therapy”.

Mazaraki et al., following Green Analytical Chemistry (GAC) principles, developed a method by combining fabric phase sorptive extraction (FPSE) with UHPLC-MS/MS to quantify six beta blockers, as atenolol, nadolol, metoprolol, oxprenolol, labetalol, and propranolol. In addition to drug monitoring, this study also aimed to quantify these drugs in case of doping. FPSE allows the time, analytes, and solvents consumption to be avoided in sample pre-treatment, which is very useful for these applications. A binary gradient allows the quantitative results to be obtained within 15 min [9].

Another field of application can be ascribed to some specific diseases or emerging disease. Mathis et al. developed a rapid method to quantify 12 metabolites to diagnose nine types of inborn metabolism dysfunctions which cause epilepsy. Through LC-MS/MS, they analyzed plasma and urine samples through a gradient LC run for a total runtime analysis of 16 min [10].

LC-MS/MS is very widely used within the field of the pharmacotoxicology thanks to its versatility, and for this reason commercial kits can often be found on the market. Thanks to older studies in TDM, using these kits it is possible to obtain a fast method transfer, maintaining the reproducibility of the results and a direct and fast method validation on-site. Furthermore, using these kits it is possible to reproduce the same method in different laboratories with the advantage of immediately comparing the results [4].

In toxicology, LC-MS or tandem mass spectrometry (MS/MS) is so frequently used because firstly it can be used for non-volatile and heat-labile compounds, unlike gas-chromatography mass spectrometry (GC-MS). Another very important and advantageous factor in the use of LC-MS/MS compared to GC-MS is that it is possible to avoid processes of derivatization of the analytes in order to make them volatile and/or analyzable by GC. This factor not only reduces the analytical variability (fewer pre-analytical steps) but contributes to reducing the time for analysis. In addition, biological samples such as blood and urine can be easily analyzed with minimal sample manipulation. It allows the classical drawbacks generally encountered during this phase to be reduced (errors related to the sample treatment and the reduction of the time). In addition, the use of high-resolution mass spectrometry (HRMS) is not so widespread, mainly because this instrument configuration is especially devoted to proteomic approaches and for qualitative purposes.

The last matrix has a vital role in detecting drugs, it is simpler to collect compared to blood and patients are more compliant. The use of these types of samples has another advantage related to the possibility of evaluating the analysis of drug metabolites to perform PK and PD studies [11]. LC-MS/MS is used in toxicology to investigate antidepressants, antipsychotics, and benzodiazepines (BDZ). Now, benzodiazepines are used and, indeed, the number of prescriptions has increased in the last few years. Another problem can be the ease with which they can be acquired on the online market [12].

LC-MS/MS, in addition to its common role for analyzing lawful drugs and illicit drugs, can also detect metabolites that come from phase I or II metabolisms. This characteristic differs from other instruments and represents the main advantage related to the use of LC-MS/MS in PK and PD studies or for pharmacotoxicological purposes [11]. In their study, Merone et al. used LC-MS/MS to develop a rapid screening method to assess more than 739 licit and illicit substances. This vast number is possible thanks to the fast gradient LC run, the polarity switching mode available on all recent MS instrumentations (to detect positive and negative ions), and the availability of the different molecular ion to daughter ion transitions for multiple reaction monitoring acquisition mode (MRM). The study started from the popularity of these substances, especially benzodiazepines, which are not used as prescribed drugs, but often as recreational substances [13].

Methling and his group examined hair to reveal drugs, including antidepressants, antipsychotics, and benzodiazepines. The selected matrix allows the “window” related to the possible drug assumption to be increased, especially related to the bioaccumulation in the hair keratin matrix. This point is further supported in the field of biological matrices analyses regarding the “time window” that can be monitored based on the type of sample being analyzed. For example, in the case of blood tests, a few hours are evaluated, while in urine tests, a few days are evaluated. It should be noted that, for the analysis of the hair, depending on the length, is possible to monitor a few months or years (completely similar for other keratin matrices). In 442 post mortem samples analyzed, 49 of 52 analytes were found. Antidepressants and antipsychotics are commonly found in post mortem toxicology due to their high prescription rates and relatively high toxicities in overdose cases. This method considers a fast run of 18 min in gradient elution mode. Thanks to the type of samples chosen (hair), the authors were able to highlight that reducing drug concentration in the different sections of the hair could represent a scaling down of the drug during the last four months. If there is a build-up of the drug, despite the above mentioned, it could coincide with the starting of therapy [14] or the start of consumption.

Campelo and coworkers detected that, in recent years, people have often resorted to antidepressants. For this reason, they developed an LC-MS/MS method with QuEChERS (Quick, Easy, Cheap, Effective, Rugged, and Safe) extraction procedure. The instrumentation condition sees a gradient run from 10 to 95% methanolic ammonium formate solution (2 mmol/L) with 0.1% formic acid. Instead, the mobile phase A was the same as B, but with an aqueous solution. In merely 8 min of analysis, it was possible to quantify the twenty most common antidepressants. The method was validated on post mortem blood, and when it was applied on real samples, the absence of antidepressants was confirmed. The limits of quantification (LOQ) were 10 ng/mL for all the analytes [1], highlighting the great sensitivity (in this matrix) of the hyphenated LC-MS/MS instrument configuration.

In PK and PD field, drug concentration is affected by absorption, distribution, metabolism, and excretion (ADME). With metabolism and its enzymes, such as cytochrome, playing an important role, the concentration and relative effects can be different for each person. For these reasons, is important to follow therapeutic drug monitoring to personalize therapy [15]. For example, when there is low quantity or low concentration of a drug, the better instrument is LC-MS/MS, thanks to its sensitivity. One example is the study conducted by Liu et al. in 2016. Because of the low absorption of Naloxone and its lower quantity compared to the other drug used in the formulation, the better method of analysis is LC-MS/MS allowing for the determination of up to three pg/mL of Naloxone [16].

Da et al. in 2018 used dried blood spot (DBS) as sample to detect the concentration of Fluoxetine by LC-MS. This drug is common in patients suffering from depression but, at the same time, it needs therapeutic drug monitoring because each patient responds in a different way due to metabolism by cytochrome. Moreover, these patients are not hospitalized, so in this way, they found a good compromise between analysis and patients’ compliance [17].

Additionally, Linder et al. emphasized the facility of analysis of dried blood spots using LC-MS [18,19]. In this study, the group compared the concentration of drugs from dried blood spots and plasma, analyzed the first one in LC-MS/MS and the second one by immunochemical methods. Starting from this study, they aimed to convert plasma and dry blood spot (DBS) because these preliminary results showed excellent correlations.

Table 1 reports the LC-MS/MS characteristics of the most recent application in the field of pharmacotoxicology and the general instrument configuration applied.

## 3. Illicit Drugs

Nowadays, illicit drugs are a real problem. In particular, these compounds can cause accidents and death on the road. Additionally, effects on the central nervous system (CNS) can be psycholeptic, psychoanaleptic, or psychodiseptic, and who consumes them can create problems related to public and social order due to pharmacological disorders such as tolerance, addiction, or dependence [3].

For these reasons, to help law enforcement, oral fluid (OF) is considered on par with plasma because the illicit drug concentrations in both are similar (or a possible correlation between the concentrations in the different matrices is known). Law enforcement can collect saliva samples non-invasively without any medical supervision, thanks to standardized procedures and specific devices developed to assure the accuracy of the analysis and reproducibility. The use of oral fluid is mainly related to military and law enforcement investigations. Over the years, tools and devices have been developed which allow even untrained personnel in the sampling phases to proceed with the collection of oral biological fluids. These samples are of primary importance as, by means of these standardized devices, they allow presumptive and preliminary screening. This type of analysis can be used, for example, in conjunction with field sobriety tests to help confirm or dispel suspicions of abuse (both alcohol and illicit substances). Other applications may include, but are not limited to, test period monitoring, post-accident evaluation, surface testing of unknown substances, etc. In particular, the use of oral fluid allows a series of advantages to be obtained that can hardly be obtained with other procedures. This sampling is rapid (samples can be taken at or near the time of incidence and results are provided on site), easy (oral screening does not require observed same-sex collection), reliable (if the procedure and screening test is properly validated, it is difficult to adulterate the sample), non-invasive (no need for medical professionals to take samples), and hygienic (if handled properly, administrators will not encounter a donor’s oral fluid).

Bassotti et al. report an example of the application of this concept in 2020. This study developed a rapid method to determine 17 different illicit drugs in OF, and performed the analysis of these analytes with a simple sample pre-treatment, named “dilute and shoot”, a run time of 12 min, and only water with 0.1% formic acid and 50:50 Acetonitrile (ACN)/Methanol (MeOH) with 0.1% formic acid as mobile phases in gradient elution [20].

Following their in-depth studies in 2022, the same group published a new method for the determination of up to 739 compounds. The considerable number includes both licit and illicit substances, from antiepileptic drugs to cannabis, from benzodiazepines to hallucinogens. This study started from a consciousness of increased use of illicit substances and the corresponding decrease in licit ones, such as tobacco or alcohol. Thanks to this method, it can be possible to perform a rapid screening test (qualitative) followed by a confirmation analysis that needs more time. A method like this is essential, especially in legal cases, if there is a possibility of intoxication or overdose by unknown substance(s). The research group used for this method blood and urine from routine clinical TDM and post mortem blood from autopsies. In this study, particular attention was paid to green chemistry; for this aim, they used MeOH for protein precipitation for blood and a solution with glucuronidase for hydrolysis reaction for urine. The LC-MS/MS method involves a run time of 18 min, during which flow changes were applied for a gradient elution [13].

A group of drugs, called new psychoactive substances (NPS), has been developed in recent years. NPSs are characterized by chemical modifications to classic drugs and pharmaceuticals. This class is not a medical drug, but it is only used to have fun, without PK study and/or toxicity information or mortality rate [12]. In 2020, the European Monitoring Centre for Drug and Drug Addiction (EMCDDA) described 46 new substances of NPSs. In Italy, the most used are synthetic cathinones (mephedrone, α-PHP, 3-MMC, eutylone), synthetic cannabinoids (JWH-122 and JWH-210), and opioids (ocfentanil, 2-methyl-AP-237 and car-fentanyl) [21].

Vaiano and his team developed an LC-MS/MS method to quantify drugs and illicit drugs in blood. For 120 NPS and 43 prescription drugs, the developed method required merely 37 min of runtime, a little bit too long if we compare with Merone et al. [13]. However, they had good sensitivity and linearity and, compared with other studies, their sample pre-treatment is easier because it is just a protein precipitation using cold acetonitrile [22].

It is vital to counter illicit drug use because they affect perception and driving skills, such as benzodiazepines (BDZ) and opioids. Another source of confusion originates from the package leaflets that create the wrong impression to the consumers [5].

Lau and his group used post mortem blood, especially femoral and heart blood, to quantify up to 30 different synthetic cathinones using LC-MS/MS. They became aware of the disproportionate use of these illicit drugs because the abovementioned were very famous in America, inducing significant amphetamine-like symptoms [23].

Sample pre-treatment consists of solid phase extraction (SPE), and thanks to a gradient LC run (mobile phase water and acetonitrile both with 0.1% formic acid) in 16 min, the analytes of interest are easily found. The LOQs were for every analyte at the concentration level of 1 ng/mL [23]. Ferrari Junior and Caldas also used the same mobile phases in their research about the determination of 79 substances, which includes 23 prescription drugs, 13 synthetic cathinones, 11 phenethylamines, 8 synthetic cannabinoids, 7 amphetamines, and 17 other psychoactive substances. The researchers used biological samples, such as blood and urine, because the first one is the most used in intoxication cases, while the second one can trace these compounds well [24].

The main difference between the two studies is Ferrari’s attention to the QuEChERS method. The group tried a pre-treatment protocol with different samples for the extraction using a different amount of water and ACN, succeeded by many procedures that ended with reconstituting with 200 µL of ACN and 0.1% formic acid. The mobile phases consist of a fast gradient elution starting by 1% of the mobile phase, up to 99% B at 10/12 min, and returning to 1% B at the end of the LC run, which is 14.1 min. Otherwise, the flow rate is 0.5 mL/min, and the temperature is 40 °C. In addition, the injection volume is assessed, optimizing at 1 µL [24] to evaluate the highest ionization efficiency in the MS source (reflecting the highest sensitivity).

Using urine as sample of analysis, Kahl and coworkers quantified drugs of abuse. It is interesting to pay attention to the methods used, because they compared LC-MS/MS to enzyme-linked immunosorbent assay (ELISA), concluding that the limit of detection was lower in the first one. Furthermore, LC-MS/MS showed better “flexibility” than the immunoassay, especially for the newest drugs, concluding that although the LC-MS/MS method needs more times than the immunoassay, the last one can be just a screening test [25].

Broecker et al. chose hair as a specimen because it carries a clear drug usage history over time, even for years. This type of choice is used in toxicology to follow the use of illicit drugs. Hair samples were collected during the autopsy and from laboratory staff. The pre-treatment consists of adding methanol/ACN/2 mM ammonium formate (25/25/50, *v*/*v*/*v*) and incubating for 18 h at 37 °C to extract analytes. The temperature was chosen based on the possible analyte decomposition. The choice of post mortem hair was made in collaboration with police because law enforcement significantly contributed to the history of a single person’s death. They followed 30 illicit drugs, mainly experiencing heroin, thanks to the presence of its metabolites and cocaine, following the decrease in the levels of crack and amphetamine [26]. Broecker, and Rubicondo and coworkers used hair to check NPS and drugs. Specifically, Rubicondo and coworkers started from research performed last year, using blood as a sample and adding seven other substances. They obtained a complete method to quantify up to 120 NPSs, 43 BDZs/antidepressants, and 6 opiates/opioids. Sample pre-treatment was easier than Broecker’s plan, because it involved only a washing step followed by protein precipitation that, unlike mobile phases, is 5 mM aqueous with formic acid and Acetonitrile 99:1, respectively. Fascinating was the application to samples of 100 people who previously completed a questionnaire about drug consumption. In some cases, the answers do not match the results from the analysis. Tradozone was the most discovered drug during analysis, followed by BDZ as flunitrazepam and diazepam. In addition, two synthetic cathinones, methylone and mephedrone, were found [5].

Baumgartner, in 2012, like his above-mentioned colleagues, used hair as sample. He focused attention on a screening test, called VMT-A, that is an immunochemical method, confirming it by LC-MS/MS. He obtained good results with VMT-A, but reported the importance of confirmation by LC-MS/MS [27].

Another matrix that can be used in LC-MS/MS is muscle, because sometimes whole blood cannot be available for routine analysis of forensic cases. In fact, from April 2019 to January 2021, in 108 cases of 800, blood was unavailable for analysis. For this reason, Hansen et al. developed a UHPLC-MS/MS method correlating blood and muscle analysis results. They considered 29 drugs and metabolites using a C18 column with a gradient analysis of 0.025% *v/v* ammonia aqueous solution and methanol. The step that differs between the two types of matrices is a homogenization step for muscle, and then the method was fully validated [28].

A different approach that can be considered in the case of lack of blood, can be skeletal tissue [29], as Orfanidis did. In their work, they first developed the method using bone, which was difficult because bone is a hard tissue that is complex to pre-treat, unlike blood. After different experiments, they found the right way to extract analytes using MeOH, ammonium hydroxide (NH_4_OH), ultrasonic bath, and centrifuge. After this sample treatment, the sample can be directly injected and resolved in LC by a gradient analysis with water and methanol, both with 0.1% formic acid, proving that bone can be a possible (and valuable) alternative matrix for forensic analysis. They detected 27 drugs that applied firstly in two cases from chronic abusers, from antidepressants to cocaine and opiates [29]. They developed two UHPLC-MS/MS methods for determining 84 drugs, licit and illicit, starting from different QuEChERS protocols. The two methods were developed because the matrices differed from blood [30] and liver [31]. This review will discuss liver protocol because it is less common. Due to a large number of analytes of interest, they decided to work with a C18 column with a total length of 150 mm. In both methods, the mobile phases were water and 0.1% formic acid and methanol with the same percentage of acid. The run can continue with a linear gradient if starting with a high water percentage to lower the retention of hydrophobic compounds [31].

Following different forensic cases, often is necessary the approach to new matrices. One of the last (and valuable) matrices used is nails [32]. As reported by Mannocchi and coworkers, nails can be useful in toxicological analysis. Nails cannot replace conventional and safer matrices, but they can monitor chronic exposure (nails are a keratinic matrix, just like hair). Their study developed a method for nails and hair in LC-MS/MS, applied in actual samples of 87 NPSs and 32 other illicit drugs, encountering success [33]. Thanks to this evidence, if we have new unconventional matrices to use in forensic cases, there are new goals to reach in analytical chemistry, both with the most significant aim of monitoring illicit drugs and subsequently decreasing illegal acts [13].

In the pharmacotoxicological field, the last few years have been fundamental for the discovery of new methods, and the biggest aim is always the reduction in time and a lower consumption of samples. These goals were found by using MALDI/MS and often there are correlations in the literature between LC-MS/MS and MALDI/MS.

Last year’s MALDI is prevailing for molecules with high molecular weight, such as nucleic acid, proteins, and microorganism such as bacteria and fungi. For this reason, MALDI coupled with Time-of-Flight (TOF) is frequently used in TDM, but often to search for genotype, gene mutations, etc. [15,34]. This represents a good advantage because MALDI does not need method optimization for the type of sample or choice of column for each experiment, and last but not least it is less time-consuming [34].

Over recent years, new methods have been validated to semi-quantify eight benzodiazepines [35] and four pyrrolidino cathinones, both in human blood [36]. In these studies, authors have highlighted analogies and differences of MALDI and LC-MS/MS in terms of sensibility and results.

Table 2 reports the main LC-MS/MS characteristics for the last recent applications.

## 4. Conclusions

The last several years have given an important impetus to the technologies and the study of LC-MS/MS, even if the type of samples is not conventionally used. Therefore, the wish is for the use of this instrument configuration of analysis to become a daily routine, because it demonstrates the required selectivity and sensitivity that performs better than other configurations, is more reliable, and is an ideal scenario for different applications. Especially in the pharmacotoxicological area, it has demonstrated great performance from different laboratories, becoming the first choice of instrument configuration for the analysis and study of death from overdose or instances of doubt. Certainly, the applications, the instrumental configurations, and the methods currently in use are very different from those that Professor Gérard Hopfgartner could perhaps have imagined, but certainly the fundamental merit lies in the fact that he has contributed to developing a new way (and a new instrumentation) to approach sensitive and selective quantitative analysis. In addition, it could be said that it has also paved the way for a new way of thinking for analytical chemists involved in this field of analysis, in which flexibility (not only of the instrumentation) is an essential requirement in order to be able to respond to the requests and needs of an ever-evolving society.

## Figures and Tables

**Table 1 molecules-28-02127-t001:** LC-MS/MS characteristics.

n. of Analytes	Type of Matrix	Column	Source	Mass Analyzer	Mass Spectrometry Technique	Ref.
9 kinase inhibitors 2 metabolites 2 antiandrogens	Human Plasma	Acquity UPLC^®^ T3 HSS C18 analytical column 2.1 × 100 mm 1.8 μm particle size	ESI+	TQD	MRM	[7]
4 antibiotics	Human Plasma	C18 column MassTox TDM Series A basic kit	ESI+	QTrap 5500 MS	MRM	[8]
6 beta blockers	Human serum Human urine	Acquity UPLC C_18_ BEH 100 × 2.1 mm, 1.7 μm	ESI	TQD MS/MS	MRM	[9]
12 metabolites	Human plasma Human urine	Supelco Discovery HS F5 HPLC column	ESI+	5500 Triple Quad MS	MRM	[10]
739 compound licit and illicit	Human blood Post mortem human blood Human urine	Restek Allure PFPP 5 µm, 60 Å, 50 × 2.1 mm	ESI	4500 QTrap Plus	MRM	[13]
52 compound BDZ Tricyclic/tetracyclic antidepressant, selective serotonin inhibitors, and others typical and atypical neuroleptics	Human hair	C18 column 150 × 2.1 mm i.d., 1.7 μm, Phenomenex	ESI+	6460 Triple Quad MS	Dynamic MRM	[14]
20 antidepressants	Post mortem human blood	Atlantis T3 150 × 3.0 mm i.d., 3.0 μm	ESI+	Triple Quad MS	MRM	[1]
Buprenorphine Norbuprenorphine Naloxone	Human plasma	Unison UK-C18 2.0 × 50 mm; 3 µm	ESI+	5500 Triple-Quad MS	MRM	[16]
Fluoxetine Norfluoxetine	Dried blood spot	Accucore C18 100 × 2.1 mm p.d. 2.6 µm	ESI+	Triple Quad MS	MRM	[17]
Carbamazepine Lamotrigine Valproic acid	Dried blood spot	Acquity BEH C18 2.1 × 100 mm 1.7 µm	ESI+	Triple Quad MS	MRM	[18,19]

**Table 2 molecules-28-02127-t002:** LC-MS/MS characteristics.

n. of Analytes	Type of Matrix	Column	Source	Mass Analyzer	Mass Spectrometry Technique	Ref.
17 drugs of abuse	Oral fluid	Hypersil PFP Gold column 50 × 2.1 mm 1.9 μm particle size	ESI+	Triple Quad MS	MRM	[20]
739 compounds both licit and illicit	Human blood Post mortem human blood Human urine	Restek Allure PFPP 5 µm, 60 Å, 50 × 2.1 mm	ESI	QTrap	MRM	[13]
120 NPSs 43 drugs	Human blood	Zorbax Eclipse Plus C18 2.1 × 100 mm, 1.8 µm, Agilent Technologies	ESI+	Triple Quad MS	MRM	[22]
120 NPSs and 49 drugs	Human hair	Zorbax Eclipse Plus C18 column 2.1 × 100 mm, 1.8 µm; Agilent Technologies	ESI+	Triple Quad MS	MRM	[5]
30 cathinones	Post mortem human blood	Poroshell 120EC-C18 column 2.1 mm × 100 mm × 2.7 μm	Jet stream-electrospray ionization+	Triple Quad MS	MRM	[23]
23 prescription drug 13 synthetic cathinones 11 phenethylamines 8 synthetic cannabinoids 7 amphetamines other 17 psychoactive substances	Post mortem human blood Post mortem human urine	Acquity UHPLC BEH C18-column 2.1 mm i.d. × 100 mm 1.7 µm particle size	ESI	Tandem Quad MS	MRM	[24]
30 drugs of abuse	Human hair	Zorbax Eclipse plus C18 2.1 mm × 100 mm 3.5 μm	ESI+	QTOF		[26]
29 drugs and metabolites	Post mortem human blood Post mortem human muscle	ACQUITY UPLC^®^ BEH C18 1.7 µm 2.1 × 50 mm	ESI+	Triple Quad Tandem MS	MRM	[28]
27 drugs, licit and illicit	Human skeletal tissue	Acquity BEH C18 column 150 × 2.1 mm i.d. 1.7 μm	ESI+/−	TQD	MRM	[29]
84 drugs of abuse and pharmaceuticals	Post mortem blood	Acquity BEH C18 column 150 × 2.1 mm i.d. 1.7 μm	ESI+/−	TQD	MRM	[30]
84 pharmaceuticals and drugs of abuse	Post mortem human liver	Acquity BEH C18 column 150 × 2.1 mm i.d. 1.7 μm	ESI+/−	TQD	MRM	[31]
87 NPS other 32 drugs of abuse	Post mortem human hair Post mortem human nails	Oasis HBL 5 µm 4.6 × 20 mm	ESI+	Triple Quad MS	MRM	[33]
88 drugs and illicit drugs	Human urine	Agilent Poroshell 120 EC-C18 3.0 × 5 mm 2.7 µm	ESI+	TripleQuad MS	Dynamic MRM	[25]
14 drugs and illicit drugs	Human hair	Synergi 4 µm POLAR-RP 80A 150 × 2.0 mm	ESI	QTrap 3200	MRM	[27]

## Data Availability

Data and information are available on request to the authors.

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
