# Peer review of "LC-MS/MS Application in Pharmacotoxicological Field: Current State and New Applications"

_molecules, 2023, doi:10.3390/molecules28052127_

Round 1
Reviewer 1 Report
In the submitted manuscript (review paper) the applications of LC-MS/MS in pharmacology and forensics are discussed. Although covering important topic and providing some interesting data the manuscript presents several flaws.
The main problem is related to the confusing structure and the poor language (e.g. lines 165-169) that requires a deep scientific revision.
In many cases English style is also poor, thus making certain sections difficult to understand e.g., lines 111-116; 129-139, 141-142, 193-194, 208-212, 252-253
In addition, there are some other issues that need to be addressed:
1) The title “LC-MS/MS application in forensic field: current state and new applications” might be a bit misleading since a part of manuscript is focused on LC MS used in pharmacology/TDM etc.
2) Very often studies are cited with lots of unnecessary data and without summing up/conclusion.
3) I suggest for the paper to be reorganized and rewritten to focus on forensics. Authors might organize revised version in following manner – 1) types of samples analyzed (biological and non-biological), 2) types of samples preparation, 3) types of LCMS instrumentation (for example there is no data on MALDI).
4) The last but not the least, there are no limitations of the LCMS listed as well as comparison to some other methods such as automated immunoassays which are sometimes the preferred method compared to the quality of results generated from LC-MS.
A paper that might provide some idea on new structure of the manuscript might be Brown et al, The current role of mass spectrometry in forensics and future prospects, 2020 https://doi.org/10.1039/D0AY01113D
Author Response
Assistant Editor, MDPI Kraków
Recently we sent you an email about major revision, additionally, we would like to kindly inform you that the article needs to be improved in length. The article is an original research manuscript, and the structure should include an Abstract, Keywords, Introduction, Materials and Methods, Results, Discussion, and Conclusions (optional) sections, with a suggested minimum word count of 4000 words. Please check the details in the following link: https://www.mdpi.com/journal/molecules/instructions. We checked the number of words in the main text of your manuscript is 3799, and we suggest you to further enrich the content of the article.
Dear Assistant Editor, the herein paper is a Review paper (and not an original research manuscript), and for this reason your suggestion about article structure cannot be adopted.
English Editing Department
One of the reviewers has suggested that your manuscript should undergo extensive English revisions. Note that extensive English editing is not included in the APC. This may not accurately reflect the English level of your manuscript but we recommend that you further check the English during revision. If the reviewer does not provide detailed comments, you may ask the assistant editor to contact the reviewers for more specific comments. We propose that you have your manuscript checked by a native English-speaking colleague, or use a paid editing service. We also offer paid editing services at https://www.mdpi.com/authors/english.
Dear English Editing Department, the herein revised paper is thoroughly checked by one of the authors that, as highlighted by the affiliation, contributed from USA.
REVIEWER #1
In the submitted manuscript (review paper) the applications of LC-MS/MS in pharmacology and forensics are discussed. Although covering important topic and providing some interesting data the manuscript presents several flaws. The main problem is related to the confusing structure and the poor language (e.g. lines 165-169) that requires a deep scientific revision. In many cases English style is also poor, thus making certain sections difficult to understand e.g., lines 111-116; 129-139, 141-142, 193-194, 208-212, 252-253
We thank the Reviewer for His/Her very helpful suggestions. In the revised version, almost all of them have been accepted and reported using Word's "track changes mode". The current structure comprise 2 main sections: drugs and illicit drugs, where the specific applications were exposed and considered. Regarding the language, the revised version is fully checked by one of the authors that, as highlighted by the affiliation, contributed from USA
In addition, there are some other issues that need to be addressed:
1) The title “LC-MS/MS application in forensic field: current state and new applications” might be a bit misleading since a part of manuscript is focused on LC MS used in pharmacology/TDM etc.
We thank the Reviewer for His/Her very helpful suggestions. In the revised version, the title was revised in “LC-MS/MS application in pharmacotoxicological field: current state and new applications”. The term “pharmacotoxicological”, in fact, cover forensic, pharmacology, and TDM fields.
2) Very often studies are cited with lots of unnecessary data and without summing up/conclusion.
As correctly highlighted, several unnecessary data was removed and the discussion was improved accordingly.
3) I suggest for the paper to be reorganized and rewritten to focus on forensics. Authors might organize revised version in following manner – 1) types of samples analyzed (biological and non-biological), 2) types of samples preparation, 3) types of LCMS instrumentation (for example there is no data on MALDI).
As previously stated, almost all of the suggestions have been accepted and reported using Word's "track changes mode". The current structure comprise 2 main sections: drugs and illicit drugs, where the specific applications were exposed and considered. Furthermore, as correctly highlighted, a short paragraph on MALDI was added, even if the application on this topic are mainly focused on molecules with high molecular weight as nucleic acid, proteins, and microorganism as bacteria, fungi. For this reason MALDI coupled with Time Of Flight (TOF) is used in TDM, but often to search genotype, gene mutation etc.
4) The last but not the least, there are no limitations of the LCMS listed as well as comparison to some other methods such as automated immunoassays which are sometimes the preferred method compared to the quality of results generated from LC-MS.
As correctly highlighted, comparison and reference to automated immunoassays was added.
A paper that might provide some idea on new structure of the manuscript might be Brown et al, The current role of mass spectrometry in forensics and future prospects, 2020 https://doi.org/10.1039/D0AY01113D
As correctly suggested, to revise the current version, the manuscript of Brown et al is also considered and added in the references.
Reviewer 2 Report
LC-MS/MS application has increased in recent years both in clinical and forensic toxicology. For this reason, this paper can collect and provide information which can be very useful. However, many amendments are required.
First of all, the english language and style should be revised. Many sentences are unclear: in particular lines 126 ,158 ("in a matrix"), 149 ("type of sample matrix", which one? Post-mortem blood as you stated later) and 277 (coworkers' nails).
Some typos must be corrected: line 176 "dilute and shoot," (dilute and shoot",), line 264 "NH4OH" (NH4OH).
Abbreviations are not always at the first mention, for example LC-MS/MS at line 54; benzodiazepines (at line 207), acetonitrile and methanol.
The authors should choose between "minutes" or "min".
Reference #5 must be checked (is an author missing?). Regarding this study, at line 241, the authors mentioned Rubicondo and Broecker's works and at line 242 they stated that: " they started from research performed last year, using blood...". This sentence is correct for the Rubicondo's paper and not for Broecker's ones. Thus this sentence must be corrected.
Table 1 and 2: column's types are not clearly readable. More space is required among the raw.
line 170-172, the authors should specify the application of oral fluid by the law enforcement.
Introduction: in my opinion, authors should compare the LC-MS/MS with other instrumentations such as GC-MS in order to describe the advantage in terms of time- and resource-consuming. A brief mention should be made for HRMS.
The authors should described how they have chosen the studies included in this paper.
Author Response
Assistant Editor, MDPI Kraków
Recently we sent you an email about major revision, additionally, we would like to kindly inform you that the article needs to be improved in length. The article is an original research manuscript, and the structure should include an Abstract, Keywords, Introduction, Materials and Methods, Results, Discussion, and Conclusions (optional) sections, with a suggested minimum word count of 4000 words. Please check the details in the following link: https://www.mdpi.com/journal/molecules/instructions. We checked the number of words in the main text of your manuscript is 3799, and we suggest you to further enrich the content of the article.
Dear Assistant Editor, the herein paper is a Review paper (and not an original research manuscript), and for this reason your suggestion about article structure cannot be adopted.
English Editing Department
One of the reviewers has suggested that your manuscript should undergo extensive English revisions. Note that extensive English editing is not included in the APC. This may not accurately reflect the English level of your manuscript but we recommend that you further check the English during revision. If the reviewer does not provide detailed comments, you may ask the assistant editor to contact the reviewers for more specific comments. We propose that you have your manuscript checked by a native English-speaking colleague, or use a paid editing service. We also offer paid editing services at https://www.mdpi.com/authors/english.
Dear English Editing Department, the herein revised paper is thoroughly checked by one of the authors that, as highlighted by the affiliation, contributed from USA.
REVIEWER #2
LC-MS/MS application has increased in recent years both in clinical and forensic toxicology. For this reason, this paper can collect and provide information which can be very useful. However, many amendments are required. First of all, the english language and style should be revised. Many sentences are unclear: in particular lines 126 ,158 ("in a matrix"), 149 ("type of sample matrix", which one? Post-mortem blood as you stated later) and 277 (coworkers' nails).
We thank the Reviewer for His/Her very helpful suggestions. The current revised version is thoroughly checked by one of the authors that, as highlighted by affiliation, contributed from USA
Some typos must be corrected: line 176 "dilute and shoot," (dilute and shoot",), line 264 "NH4OH" (NH4OH).
As correctly suggested, the manuscript was entirely checked for typos and grammar errors.
Abbreviations are not always at the first mention, for example LC-MS/MS at line 54; benzodiazepines (at line 207), acetonitrile and methanol.
As correctly suggested, at the end of the paper was added a table with all the abbreviations used.
The authors should choose between "minutes" or "min".
As suggested, the entire manuscript was checked and “minute” was selected.
Reference #5 must be checked (is an author missing?). Regarding this study, at line 241, the authors mentioned Rubicondo and Broecker's works and at line 242 they stated that: " they started from research performed last year, using blood...". This sentence is correct for the Rubicondo's paper and not for Broecker's ones. Thus this sentence must be corrected.
As suggested, this point was checked and revised.
Table 1 and 2: column's types are not clearly readable. More space is required among the raw.
Revised accordingly.
line 170-172, the authors should specify the application of oral fluid by the law enforcement.
Revised accordingly.
Introduction: in my opinion, authors should compare the LC-MS/MS with other instrumentations such as GC-MS in order to describe the advantage in terms of time- and resource-consuming. A brief mention should be made for HRMS.
As correctly highlighted, in the revised version the sentences “In toxicology, LC-MS or tandem mass spectrometry (MS/MS) is so much used because firstly can be used for non-volatile and heat-labile compounds, unlike Gas-chromatography mass spectrometry (GC-MS). Another very important and advantageous factor in the use of LC-MS/MS compared to GC-MS is that it is possible to avoid processes of derivatization of the analytes in order to make them volatile and/or analyzable by GC. This factor not only reduces the analytical variability (fewer pre-analytical steps), but contributes to reducing the time for analysis. In addition, biological samples such as blood and urine can be easily analyzed with a minimal sample manipulation. This allow reducing also the classical drawbacks generally encountered during this phase (errors related to the sample treatment and the reduction of the time).” Were reported. Additionally, the focus of the present review is the LC-MS/MS configuration.
The use of HRMS was also added as short sentence especially related to the use of this instrument for qualitative purposes or for proteomics approaches.
The authors should described how they have chosen the studies included in this paper.
Added in the introduction section as correctly suggested. Specifically the sentence “All references herein considered cover the last 3 years, except for some specific and peculiar applications for which some more dated but still recent articles have been considered” was added.
Round 2
Reviewer 1 Report
I wish to thank the authors for accepting suggestions. The paper has been significantly improved.
Reviewer 2 Report
The paper was amended as required.